# VALLEY: VIDEO ASSISTANT WITH LARGE LANGUAGE MODEL ENHANCED ABILITY

## ABSTRACT

Large language models (LLMs), with their remarkable conversational capabilities, have demonstrated impressive performance across various applications and have emerged as formidable AI assistants. In view of this, it raises an intuitive question: *Can we harness the power of LLMs to build multimodal AI assistants for visual applications*? Recently, several multi-modal models have been developed for this purpose. They typically pre-train an adaptation module to align the semantics of the vision encoder and language model, followed by fine-tuning on instruction-following data. However, despite the success of this pipeline in image and language understanding, its effectiveness in joint *video and language understanding* has not been widely explored. In this paper, we aim to develop a novel multi-modal foundation model capable of comprehending video, image, and language within a general framework. To achieve this goal, we introduce **Valley**, a **V**ideo **A**ssistant with **L**arge **L**anguage model **E**nhanced abilit**Y**. The Valley consists of a LLM, a temporal modeling module, a visual encoder, and a simple projection module designed to bridge visual and textual modes. To empower Valley with video comprehension and instruction-following capabilities, we construct a video instruction dataset and adopt a two-stage tuning procedure to train it. Specifically, we employ ChatGPT to facilitate the construction of task-oriented conversation data encompassing various tasks, including multi-shot captions, long video descriptions, action recognition, causal relationship inference, etc. Subsequently, we adopt a pre-training-then-instructions-tuned pipeline to align visual and textual modalities and improve the instruction-following capability of Valley. Qualitative and qualitative experiments demonstrate that Valley has the potential to function as a highly effective video assistant that can make complex video understanding scenarios easy. Our Code, data, and models will be open-sourced.

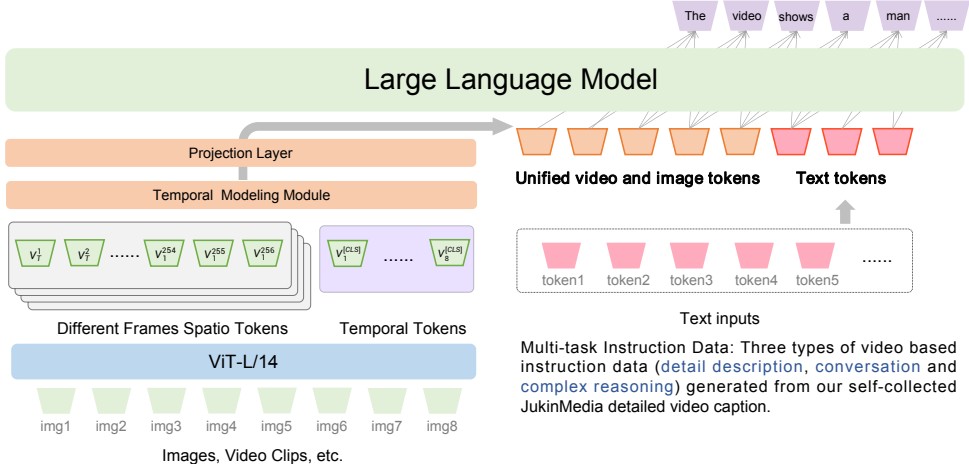

Figure 1: Valley architecture.

# 1 INTRODUCTION

The rapid growth of video applications and data has created a pressing need for automated technology to analyze and comprehend video content. This is particularly important for applications such as video surveillance, content-based video retrieval, and video summarization. However, existing video understanding models are often task-specific and lack a comprehensive capability of handling diverse tasks. Thus, it is imperative to develop more comprehensive and general video understanding models, which represent a crucial research direction for video understanding.

Large language models (LLMs) such as ChatGPT (Ouyang et al., 2022), PaLM (Chowdhery et al., 2022), and LLaMA (Touvron et al., 2023) have demonstrated their impressive abilities in understanding and following user intentions and instructions. These language models are able to learn knowledge from large-scale text corpus well and can tackle a wide range of challenges, such as text generation, summarization, translation, and question-answering. Additionally, one of the most significant changes brought by LLMs is the ability to support conversational interactions like human beings. By enabling natural and intuitive conversations, LLMs have paved the way for more seamless interactions between humans and computers. One of the most distinguished examples of such work is ChatGPT, which has become an indispensable aspect of various applications, including customer service, healthcare, and e-commerce, commonly referred to as AI assistants.

To this end, it naturally begs the question: *Can we leverage the powerful LLMs to better fuse visual and language modalities together and create a multi-modal AI assistant*? Numerous recent studies have made notable advancements in this direction. In general, these works can be grouped into two main categories. The first category utilizes Q-Former from BLIP-2 (Li et al., 2023b) to align visual and textual modalities, such as InstructBLIP (Dai et al., 2023), Otter (Li et al., 2023a) and Mini-GPT4 (Zhu et al., 2023a). The second category involves the use of a simple projection layer to achieve modality alignment, such as LLaVA (Liu et al., 2023). There also exists some attempts to extend such multi-modal works to video understanding. For example, VideoChat (Li et al., 2023c) focuses on integrating video foundation models and LLMs via a learnable neural interface and Video-LLaMA(Zhang et al., 2023a) proposes a Video Q-former to capture the temporal changes in visual scenes and audio-visual signals following Mini-GPT4 (Zhu et al., 2023a)

In this work, we endeavor to create multimodal AI assistants from multi-modal instruction data collection and multi-modal foundation model building. First, we collect 100k video samples and organize them into instruction conversation data with the assistance of ChatGPT. The dataset encompasses various types of tasks, including video detailed descriptions, conversations around video, as well as complex reasoning on video. Then, we present Valley, which employs a simple projection module as the bridge between video, image, and language modalities. For improving visual comprehension, we adopt the ViT-L/14 (Dosovitskiy et al., 2021) of CLIP (Radford et al., 2021) as the vision encoder and explore three spatio-temporal modeling strategies in the temporal modeling module to generate unified visual tokens for LLMs. Furthermore, we adopt a two-stage training strategy. During the pre-training stage, we exclusively train the projection module to enable the LLM to comprehend visual data. Following this, in the end-to-end training stage, we train both the projection module and LLM together, ensuring that Valley responds aptly in accordance with instructions. In a nutshell, our key contributions are summarized as follows:

- We propose *Valley*, a multi-modal foundation model with general comprehension and analysis ability of video, image, and language that could be a video assistant capable of engaging in conversations.

- We collect a large multi-modal instruction-following dataset with the assistance of ChatGPT, which focuses on video understanding and comprises diverse types of tasks, including detailed description, conversation, and complex reasoning. This dataset eliminates data noise caused by the addition of object information from error predictions of other basic vision models.

- Extensive experiments on visual question answering and caption benchmarks have demonstrated that our proposed Valley reaches optimal performance and powerful zero-shot capability. And it also has ability of few-shot learning without special training. Practical conversation cases show that Valley's generated content exhibits a reduced tendency for hallucinations in comparison to other analogous models.

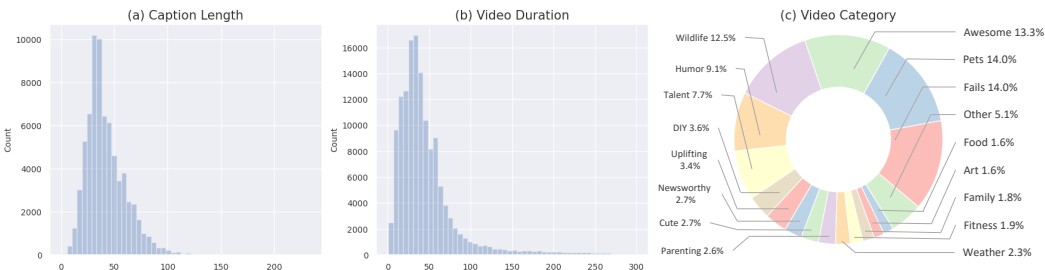

Figure 2: Distribution of video description length, video duration and video type in our self-collected 100k videos from JukinMedia website

## 2 RELATED WORK

**Large Language Models.** Large language models (LLMs) have achieved tremendous success in the field of natural language processing (Chowdhery et al., 2022; Ouyang et al., 2022; Hoffmann et al., 2022), with its excellent language understanding and reasoning abilities. LLMs can handle various complex tasks by comprehending prompts in a few-shot or zero-shot manner and have thus garnered widespread attention. Further, the development of a series of open-sourced LLMs, including LLaMA (Touvron et al., 2023), GLM (Zeng et al., 2022), and BLOOM (Scao et al., 2022), has fueled interest in this field and inspired a number of works such as Alpaca (Taori et al., 2023), Vicuna (Chiang et al., 2023), and ChatGLM. Due to their large number of parameters, LLMs can not only acquire notable task transfer generalization ability but also complete actual tasks conversationally by aligning with human language instructions and preferences. Based on these, our goal is to extend the capability of LLMs to video-grounded conversation scenarios.

**LLMs for Multimodal Understanding.** As LLMs have demonstrated strong general capabilities in linguistic tasks, enabling LLMs to understand multimodal content has been increasingly studied. Existing methods can be divided into two technical routes, one is to employ LLMs as schedulers to schedule the existing multimodal models, and the other is to train a multimodal model based on LLMs. After receiving user instructions and the functions of each foundation model, the former treats the LLMs as a controller to call corresponding models step by step and integrates the output content of each model to generate results (Wu et al., 2023; Shen et al., 2023; Yang et al., 2023). For example, HuggingGPT (Shen et al., 2023) utilize the ChatGPT to select appropriate models in Hugging Face[1] according to their function description and summarizes their execution results. The latter equips LLMs with auxiliary modules to help them understand multimodal contents through end-to-end training (Li et al., 2023c; Zhu et al., 2023a; Zhang et al., 2023a; Liu et al., 2023; Su et al., 2023; Dai et al., 2023). For instance, LLaVA (Liu et al., 2023) and MiniGPT-4 (Zhu et al., 2023a) connected LLaMA (Touvron et al., 2023) with a visual encoder through a projection layer, endowing it the ability to understand images. Video-LLaMA (Zhang et al., 2023a) empowered LLaMA (Touvron et al., 2023) with visual and audio information via Q-Former to endow it with video-grounded conversation ability.

## 3 DATA COLLECTION

We witness some open source works that collect video based instruction data, such as VideoChat(Li et al., 2023c), Video-ChatGPT(Muhammad Maaz & Khan, 2023). The work VideoChat use Web-vid(Bain et al., 2021) while Video-ChatGPT using ActivityNet(Fabian Caba Heilbron & Niebles, 2015) to build their instruction dataset. Both have drawbacks as source of instruction data generation. Video captions in Webvid are too short to contain much information for ChatGPT to generate high quality instruction data, and captions in ActivityNet are more focous on the human activities. To rich more scene and object information, they use some dense object caption generation methods such as GRIT(Wu et al., 2022) and Tag2Text(Huang et al., 2023) to provide object information in video when constructing instruction dataset. However, this will lead to a reduction in quality of the generated

---

[1]https://huggingface.co/models

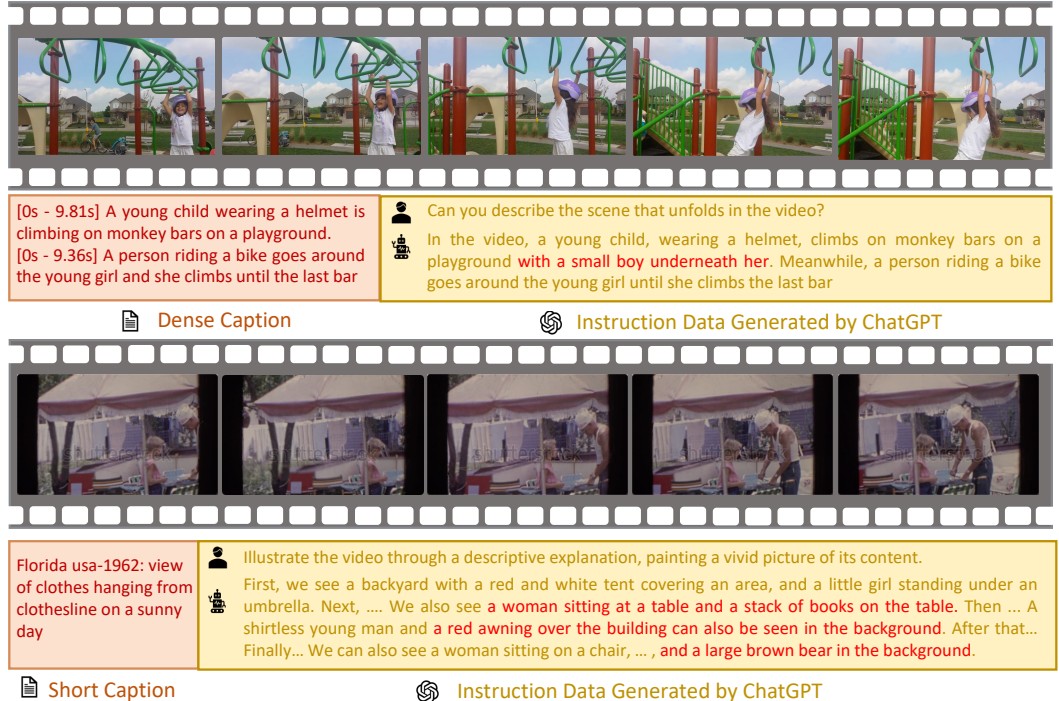

Figure 3: Two examples of hallucinations in instruction data generated by dense captions (Video-ChatGPT) and short captions (VideoChat). The red text is the illusion part.

data, leading to strong hallucinations in the trained multi-modal large language model. Two illusion examples are shown in Figure 3.

To solve this problem, we find a video website called Jukinmedia[2] that provides videos with diverse categories and wide detailed descriptions. We collect 100k videos from this website which are used as the source for instruction data generation. Figure 2 shows the distribution of video description length, video duration and category among the 100k videos we collected. We find that the average video description length is about 40 words, and the longest descriptions are greater than 100, which shows excellent detailedness in video descriptions. The average video duration is about 40 seconds, and the maximum duration is more than 5 minutes. We build a wide collection of videos, with a total of 26 categories, such as "Wildlife", "Humor", .etc.

We refer to the prompts that generate instruction data in LLaVA(Liu et al., 2023), and design three different types prompt (detail description, conversation and complex reasoning) for building our instruction dataset based on these detailed video-text pairs. For prompt details, please see Appendix A.1 . Figure 4 shows an example of our ChatGPT assisted instruction data. The instruction data constructed in this way will not bring about the illusion of object level and also contain complete video information.

# 4    APPROACH

In this section, we first introduce the overall architecture of the proposed Valley model.Then we will introduce the three structural designs of the temporal modeling module in detail. And finally we present our two-stages training procedure.

---

[2]https://www.jukinmedia.com/

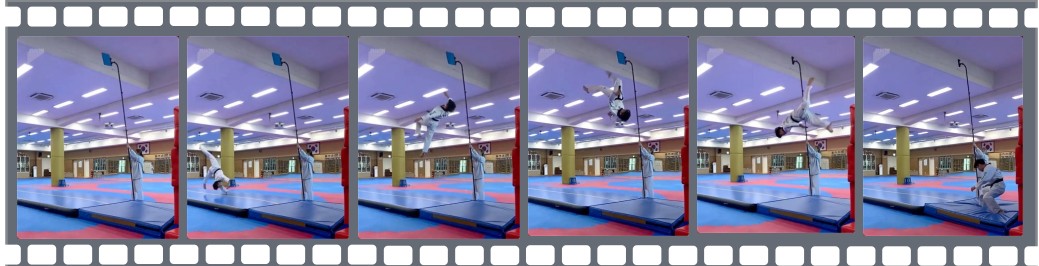

This guy did an amazing martial arts stunt. He jumped on the mat and did a backflip. At the same time, a volunteer was standing at the end of a mat holding a stick with a board at the top at the height of four meters. He amazingly kicked the board and landed safely on the mat.

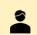 Illustrate the video through a descriptive explanation, painting a vivid picture of its content.

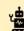 In this impressive video, a skilled individual showcases their martial arts prowess with a jaw-dropping stunt. The video begins with the person standing on a mat, poised for action. With a burst of energy, they launch themselves into the air, executing a flawless backflip mid-flight. Simultaneously, a volunteer positioned at the end of the mat holds a stick with a board balanced on top, an astonishing four meters above the ground. With incredible precision and agility, the individual executes a powerful kick, striking the board with remarkable accuracy. The force of the kick propels the board into the air, showcasing the individual's strength and skill. As the board soars, the person gracefully descends, landing safely back on the mat. This awe-inspiring display of martial arts mastery demonstrates the individual's exceptional athleticism, timing, and control.

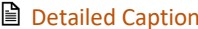 Detailed Caption

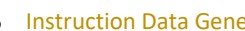 Instruction Data Generated by ChatGPT

Figure 4: An example utilizing our self-collected detailed caption generated instruction data.

## 4.1 OVERALL ARCHITECTURE

In order to allow pre-trained LLM to understand videos and adapt videos of different lengths together with individual images, we add a temporal modeling module to the vision encoder to aggregate each frame's grid features as unified vision tokens, while keeping the rest structures the same with LLaVA using a simple yet effective projection layer to connect the vision tokens to LLM. We choose Stable-Vicuna as the language interface since it exhibits superior multilingual chat abilities. The overall architecture is shown in Figure 1.

We input a video $\mathbf{v}$ and sample T frames by 0.5 FPS(1 picture per 2 seconds), which can be denoted as $\mathbf{v} = [v_1, v_2, \cdots, v_T]$. Each image obtains visual features through the pre-trained CLIP visual encoder (ViT-L/14), denoted as $V_T = \text{ViT}(v_T)$. Each feature contains 256 spatio patch features and 1 global feature ("[CLS]" token), denoted as:

$$V_T = [V_T^{[CLS]}, V_T^1, V_T^2, \cdots, V_T^{256}].$$

We use a temporal modeling module(Section 4.2) to aggregate spatial patch features of T frames in the time dimension, denoted as:

$$\hat{V} = \text{TEMPORALMOULE}([V_1, V_2, \cdots, V_T]).$$

In order to alleviate the vanishing of global temporal features caused by spatial feature aggregating, we obtain the representation $Z_V$ of the entire video by concatenating patch features after temporal modeling module and global features of T frames, the mathematical form is expressed as follows

$$Z_V = [\bar{V} \oplus V_1^{[CLS]} \oplus V_2^{[CLS]} \oplus \cdots V_t^{[CLS]}],$$

where $\oplus$ means concatenation.

We also consider a projection layer to connect visual features into the word embedding space, since it already shows its effectiveness in LLaVA. We utilize a trainable matrix to transform the video patch features and global features into the language feature space, which has the same dimension. Finally, the projected visual features and text embedding are input into LLM for response generation.

$$\hat{Z}_V = \text{LLM}(\text{Projector}(Z_V)).$$

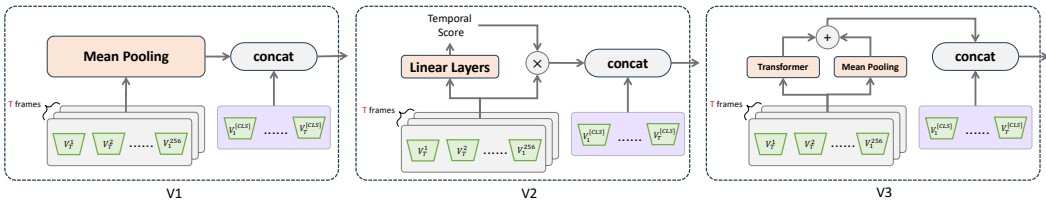

Figure 5: Illustration to three types of temporal modeling modules

## 4.2 TEMPORAL MODELING MODULE

We proposed three structures (denoted as v1,v2 and v3) to aggregate representation of spatial tokens in the time dimension, which are shown in Figure 5. For the v1 structure, we use an average pooling layer to aggregate spatial token representations. The mathematical expression is as follows:

$$\hat{V}^i = \text{AVGPOOLING}\big([V_1^i, V_2^i, \cdots, V_T^i]\big),$$

which $i$ is index of spatial token. However, v1 will cause confusion in the time dimension as it averages features in different time stamps. In order to let the model know the importance of each frame of the video to the result,we proposed v2 structure. v2 structure is introduced a learnable linear layer to learn the temporal importance score of a certain frame, and finally perform a weighted average by the scores. Expressed as follows:

$$\hat{V} = \text{Linear}\big([V_1, V_2, \cdots, V_T]\big) \cdot [V_1, V_2, \cdots, V_T].$$

In addition, we also propose a third structure v3 to model the temporal variation representation of these spatial tokens. We input spatial tokens into one layer transformer encoder, and take the representation of the last time stamp output as the variation feature of the spatial token in time information. Then we add this feature representing temporal information to average pooling feature, the mathematical representation is as follows:

$$\hat{V}^i = \text{TRANSFORMER}\big([V_1^i, V_2^i, \cdots, V_T^i]\big) + \text{AVGPOOLING}\big([V_1^i, V_2^i, \cdots, V_T^i]\big),$$

which $i$ is index of spatial token.

## 4.3 TRAINING

Inspired by LLaVA, we adopt a two-stage training framework. The first stage pre-trains the projection layer for feature alignment, and the second stage fine-tunes the language model and projection layer.

Valley supports the input of any number of images, so in the pre-training phase, we use image-text pairs and video-text pairs for pre-training. The pre-training data includes 595K CC3M image-text pairs provided by LLaVA and 702K WebVid2M (Bain et al., 2021) video-text pairs filtered by us refer to the filtering method in LLaVA. Both images and videos are input into the model in a unified way, and the prompt is as follows:

$\#\#\#\, X_{\text{system message}}$
$\#\#\#\,$ Human: $X_{\text{instruction}} < \text{patch}_1 > ... < \text{patch}_{256} >< \text{frame}_1 > ... < \text{frame}_T >$
$\#\#\#\,$ Assistant:

If a single image is input, the number of frames is 1. The image-text pair and video-text pair are constructed as a single-round dialogue, using various questions to inquire about the video content and answering with the corresponding caption.

As introduced in Section 3, we construct a 73k video based instruction dataset that consists of 37k conversations pairs, 26k complex reasoning QA pairs and 10k detail description instruction pairs. In order to enhance the ability to describe visual content in detail, we also gather 150k image instruction data from LLaVA, and 11K video instruction data from VideoChat. We use the total 234k video and image based instruction data to perform second stage fine-tuning, which freezing the ViT weight and training projection layer and the full-parameter LLM.

# 5 EXPERIMENTS

We evaluate Valley on 7 diverse datasets across two domains: MSVD-QA (Chen & Dolan, 2011), MSRVTT-QA (Xu et al., 2016), ActivityNet-QA (Yu et al., 2019), Video-ChatGPT benchmark (Muhammad Maaz & Khan, 2023) as video understanding benchmarks, and ScienceQA (Lu et al., 2022a), MemeCap (Hwang & Shwartz, 2023) as image understanding benchmarks. All the experiments are carried out in zero/few-shot manners.

MSVD-QA, MSRVTT-QA, and ActivityNet-QA are video question answering datasets. We assess the accuracy of the model's predictions and rate Valley's output on a 5-point scale using ChatGPT, the prompt using to evaluate is shown in Appendix A.1.1. Video-ChatGPT benchmark introduces dense video description and video question-answering tasks to evaluate the model's generation capability by scoring the model's output using ChatGPT in 5 different aspects. We choose FrozenBiLM(Yang et al., 2022), VideoChat(Li et al., 2023c), LLaMA Adapter(Zhang et al., 2023b), Video LLaMA(Zhang et al., 2023a) and Video-ChatGPT(Muhammad Maaz & Khan, 2023) as the baseline for video tasks.

ScienceQA is a multimodal question answering dataset about natural, social, and language science in elementary and high school science curricula. We assemble the textual and image context as Valley's input and adopt accuracy as the metric. MemeCap is a multimodal content analysis task and requires the model to generate an analysis of the underlying meaning based on the given title and image of posts. We use Bert F1 Score as an evaluation metric, which can reflect the semantic similarity between two texts. For image tasks we choose Flamingo(Alayrac et al., 2022) and MiniGPT4(Zhu et al., 2023b) as baselines. In particular, in ScienceQA task we choose GPT-3.5(Lu et al., 2022b) as the baseline.

## 5.1 SETTINGS

We employ the Stable-Vicuna (Chiang et al., 2023) as the LLM backbone and the pre-trained ViT-L/14 (Dosovitskiy et al., 2021) from CLIP (Radford et al., 2021) to encode videos and images. We first pre-train Valley for one epoch with a learning rate of 2e-3 and then fine-tune the model for three epochs with a learning rate of 2e-5 on the instruction dataset. All the experiments are conducted on $8 \times$ A100 80G GPUs.

## 5.2 QUANTITATIVE ANALYSIS

### 5.2.1 ZERO-SHOT VIDEO QUESTION ANSWER

Table 1 shows the experimental results of zero-shot inference on three video question answering benchmarks for the three different temporal modeling structures we proposed. Videos' average duration in three benchmark are different, the duration in msvd, msrvtt and activityNet increase in order. Overall, our proposed model Valley reaches SOTA on all three benchmarks. Specifically, We observe that valley-v3 has better performance on longer video understanding, reaching SOTA on both MSRVTT-QA and AcitvityNet-QA respectively. And valley-v1 has better performance on short video understanding which is SOTA on MSVD-QA benchmark.

| Method | MSVD-QA | | MSRVTT-QA | | ActivityNet-QA | |
|---|---|---|---|---|---|---|
| | Accuracy | Score | Accuracy | Score | Accuracy | Score |
| FrozenBiLM | 32.2 | – | 16.8 | – | 24.7 | – |
| VideoChat | 56.3 | 2.8 | 45.0 | 2.5 | 26.5 | 2.2 |
| LLaMA Adapter | 54.9 | 3.1 | 43.8 | 2.7 | 34.2 | 2.7 |
| Video LLaMA | 51.6 | 2.5 | 29.6 | 1.8 | 12.4 | 1.1 |
| Video-ChatGPT | 64.9 | 3.3 | 49.3 | 2.8 | 35.2 | 2.7 |
| Valley-v1 | **65.4** | **3.4** | 45.7 | 2.5 | 42.9 | 3.0 |
| Valley-v2 | 59.1 | **3.4** | 49.9 | **3.0** | 32.5 | 2.6 |
| Valley-v3 | 60.5 | 3.3 | **51.1** | 2.9 | **45.1** | **3.2** |

Table 1: Zero-shot video QA result on MSVD-QA, MSRVTT-QA and ActivityNet-QA

### 5.2.2 VIDEO-BASED TEXT GENERATION PERFORMANCE BENCHMARKING

In order to verify the performance of the text generated by valley in 5 aspects: correctness(**COR**), detail orientation(**DO**), contextual understanding(**CU**), temporal understanding(**TU**) and consistency(**CON**), we do experiments on a video-based text generation performance benchmark provided by the Video-ChatGPT. The results are shown in Table 2. We find that valley-v3 achieve the best performance in four aspects (COR, CU,TU,CON). This is consistent with the result in Table 1, because this benchmark is built on ActivityNet videos, and valley-v3 has better performance on understanding long videos. From the perspective of suppressing hallucinations, the data we constructed can indeed make Valley do a better job in generating text correctly. In the aspect of detail orientation, valley is not as good as Video-ChatGPT. This is because the instruction data we constructed loses some detailed descriptions of objects in order to ensure the correctness of data.

| Method | COR | DO | CU | TU | CON |
|--------|-----|-----|-----|-----|-----|
| VideoChat | 2.23 | 2.50 | 2.53 | 1.94 | 2.24 |
| LLaMA Adapter | 2.03 | 2.32 | 2.30 | 1.98 | 2.15 |
| Video LLaMA | 1.96 | 2.18 | 2.16 | 1.82 | 1.79 |
| Video-ChatGPT | 2.40 | **2.52** | 2.62 | 1.98 | 2.37 |
| Valley-v1 | 2.06 | 2.42 | 2.74 | 1.83 | 2.41 |
| Valley-v2 | 2.35 | 2.13 | 2.85 | 1.99 | 2.17 |
| Valley-v3 | **2.43** | 2.13 | **2.86** | **2.04** | **2.45** |

Table 2: Results on video-based text generation benchmark provided by Video-ChatGPT.

| Method | Setup | BERT-F1 |
|--------|-------|---------|
| Flamingo | zero-shot | 65.51 |
| | zero-shot COT | 58.23 |
| MiniGPT4 | zero-shot | 65.81 |
| | zero-shot COT | 68.45 |
| Valley | zero-shot | 84.82 |
| | zero-shot COT | **85.26** |

Table 3: Results on Meme-Cap, the image metaphor understanding benchmark.

### 5.2.3 SCIENCEQA AND MEMECAP

In addition to video understanding, we explore valley's chain-of-thought and few-shot capabilities on the Science-QA dataset. It can be seen from the experimental results in Table 4 that valley shows a certain one-shot inference ability and performs slightly better than zero-shot. We also notice that when using chain-of-thought (add "you need to think step by step" to system prompt), the performance of valley is comparable to GPT-3.5, while the latter has far more parameters than the former. Valley is even better than GPT-3.5 in some aspects, such as G7-12, SOC and TXT.

| Method | Subject | | | Context Modality | | | Grade | | Average |
|--------|-----|-----|-----|-----|-----|-----|-----|-----|---------|
| | NAT | SOC | LAN | TXT | IMG | NO | G1-6 | G7-12 | |
| Human | 90.23 | 84.97 | 87.48 | 89.60 | 87.50 | 88.10 | 91.59 | 82.42 | 88.40 |
| GPT-3.5 | 74.64 | 69.74 | 76.00 | 74.44 | 67.28 | 77.42 | 76.80 | 68.89 | 73.97 |
| $Valley_{ZeroShot}$ | 69.89 | 71.54 | 69.90 | 73.00 | 67.22 | 72.22 | 72.32 | 66.51 | 70.24 |
| $Valley_{ZeroShot+COT}$ | 71.00 | 72.32 | 74.72 | 76.90 | 66.98 | 80.55 | 73.49 | 70.01 | 72.25 |
| $Valley_{OneShot}$ | 71.53 | 63.66 | 74.54 | 75.83 | 64.70 | 83.33 | 71.51 | 69.14 | 70.66 |

Table 4: Results (accuracy %) on Science QA dataset. Question classes: NAT = natural science, SOC = social science, LAN = language science, TXT = text context, IMG = image context, NO = no context, G1-6 = grades 1-6, G7-12 = grades 7-12.

In terms of image metaphor understanding, the results generated by Valley are semantically closer to the standard answer than those generated by MiniGPT4 and Flamingo, the result is shown in Table 3. We show some cases in Appendix B.2.

### 5.3 QUALITATIVE ANALYSIS

In this section, we present some cases to demonstrate the superior video understanding and instruction following capabilities of Valley, as shown in Figure 6-7. More cases are shown in appendix B.1.

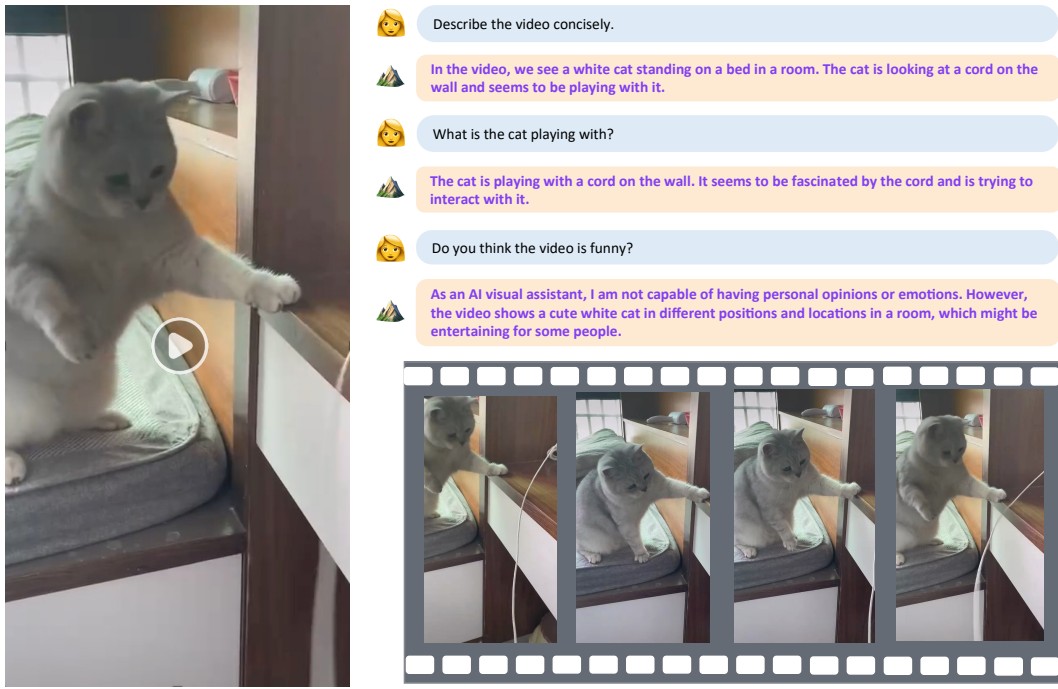

Figure 6: Video description, funny things identification, and recognition of interactions of the objects in the video.

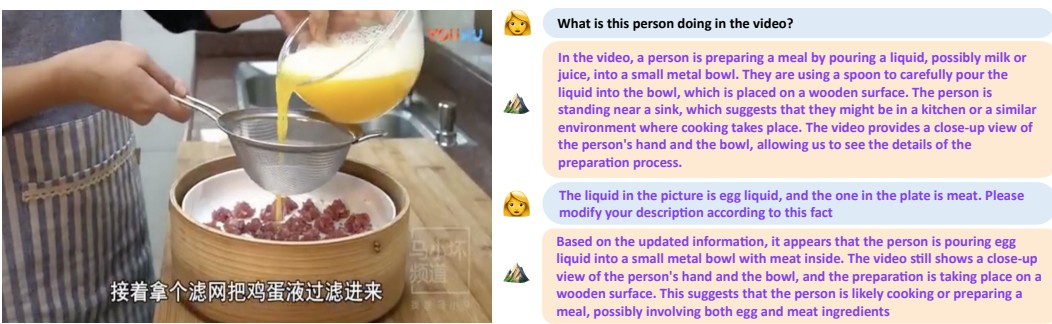

Figure 7: Image question answering and factual description with fixing mistakes.

# 6 LIMITATION

Our current Valley relies on video and language input, however, incorporating audio input will present a significant opportunity to enhance the overall user experience. On the other hand, the Valley still has some shortcomings in understanding multiple language. To tackle this challenge, we plan to construct more multilingual multi-modal data to iterate the model and improve the understanding ability of multiple language in future research.

# 7 CONCLUSION

The objective of our work is to construct a foundation model that is capable of perceiving video, image, and language in a multi-modal manner. To address this issue, we propose a framework called Valley, which stands for Video Assistant With Large Language Model Enhanced Ability. In terms of data collection, we collect 100k videos with detail caption from website and generate a multi-task instruction-following video dataset with higher quality assisted by ChatGPT. Ultimately, our goal is to create a more intuitive, personalized, and human-like interaction between humans and machines.

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

# A  PROMPT

## A.1  PROPMT FOR INSTRUCTION DATA GENERATION

The prompts we use to generate three types of instruction data are shown in table 5, table 6 and table 7 respectively. Regarding the detailed description type and complex reasoning type data, we will give a manually written example and let ChatGPT generate it using few-shot learning.

---

**Prompt for Detail Description**

**System**: You are an intelligent assistant that can understand video information through text descriptions. You can understand the overall content of the video from the title of the video, the caption of the video. Please describe the video you saw through the information given above. Don't mention the title in your description. Don't copy the original caption. Do not separately describe which objects are included in the video. It is necessary to integrate object information into your description through adjectives or attributive clauses. This description should be between 150 and 200 words.

**User**:
[title] Guy Scratches Head After Landing Perfect Bowling Strike [Caption] This guy scratched his head in confusion after making a mind-blowing attempt at bowling. He swung his hand to release the ball but accidentally tossed it towards the gutter. However, it spun and turned at the side edges of the lane and then struck all pins in one go.

**ChatGPT**: In the video, we see a man wearing a maroon shirt and shorts standing in a bowling alley, holding a bowling ball. First, he swings his hand to release the ball but accidentally tosses it towards the gutter. Next, the ball spins and turns at the side edges of the lane, seemingly heading towards the gutter, but suddenly changes direction and heads towards the pins. · · ·

**User**:
[title] · · · [Caption] · · ·

---

Table 5: The Prompt for generating instruction data of detail description with few shot learning.

---

**Prompt for Conversation**

**System**: The task is to generate a conversation between two people. One person is watching at a video, and the other person is asking questions about the video. What they see will be provided below with some sentences. Include at least one complex question that requires reasoning and thinking. Only include the questions with certain answers that one can answer with the provided sentences. Make the QA sound like they are seeing the video. Do not use any words that may sound like looking at text instead of images, like "specify", "mention", "description", "text", "provided information", "sentence", "caption", etc. Use words like "see", "look", "view", "show", etc. Format each QA pair in a single line as a JSON dictionary. Do not include any other explanation.

**User**:
[title] · · · [Caption] · · ·

---

Table 6: The Prompt for generating instruction data of conversation.

### A.1.1  PROMPT FOR EVALUATING QA TASK

The prompt that we used ChatGPT to assist in the evaluation in the experiment is shown in the table 8. This prompt will allow the AI to judge the accuracy of the model prediction results and score the consistency of the answer.

# B  MORE CASES

## B.1  GENERAL CASES

Figure 8 shows some examples of valley's capabilities in causal reasoning and object identification.

**Prompt for Detail Description**

**System**: You are an AI visual assistant that can analyze a single video. You receive a title of this video and a caption of this video, each describing the same video you are observing. The task is to use the provided title and caption, create a plausible question about the video, and provide the answer in detail.Create complex questions beyond describing the scene.To answer such questions, one should require first understanding the visual content, then based on background knowledge or reasoning, either explain why things are happening that way, or provide guides and help to user's request. Make the question challenging by not including the visual content details in the question so that the user needs to reason about that first. When using the information from the caption, directly explain the scene, and do not mention that the information source is the caption. Always answer as if you are directly looking at the video.

**User**:

[title] Woman Pranks Sister by Covering Inside of Her Whole House in Aluminium Foil

[Caption] This woman had gone on a vacation. However, she was shocked when she entered her house on returning. Her sister had covered her whole house with aluminum foil from inside to prank her. She laughed uncontrollably as she saw everything covered in the foil.

**ChatGPT**:

{

"question": "Given the sister's initial reaction of uncontrollable laughter upon discovering the prank, how might this prank affect their relationship in the long run, considering psychological and social aspects?",

"answer": " From a psychological perspective, humor plays a significant role in maintaining healthy relationships. The sister's reaction of laughter suggests that she found the prank amusing, which could enhance their bond. Shared laughter can increase feelings of intimacy and social cohesion, indicating that the prank may have strengthened their relationship. ⋯ "

}

**User**:

[title] ⋯ [Caption] ⋯

Table 7: The Prompt for generating instruction data of complex reasoning with few-shot learning.

## B.2 MEME-CAP CASES

Figure 9 shows 2 cases about valley's understanding of picture metaphors.

**Prompt for Detail Description**

**System**: You are an intelligent chatbot designed for evaluating the correctness of generative outputs for question-answer pairs.
Your task is to compare the predicted answer with the correct answer and determine if they match meaningfully. Here's how you can accomplish the task:
− − − − −−
## INSTRUCTIONS:
- Focus on the meaningful match between the predicted answer and the correct answer.
- Consider synonyms or paraphrases as valid matches.
- Evaluate the correctness of the prediction compared to the answer.
**User**: Please evaluate the following video-based question-answer pair:

Question: {question}
Correct Answer: {answer}
Predicted Answer: {pred}

Provide your evaluation only as a yes/no and score where the score is an integer value between 0 and 5, with 5 indicating the highest meaningful match.
Please generate the response in the form of a Python dictionary string with keys 'pred' and 'score', where value of 'pred' is a string of 'yes' or 'no' and value of 'score' is in INTEGER, not STRING.
DO NOT PROVIDE ANY OTHER OUTPUT TEXT OR EXPLANATION. Only provide the Python dictionary string.
For example, your response should look like this: {'pred': 'yes', 'score': 4.8}."rom inside to prank her. She laughed uncontrollably as she saw everything covered in the foil.

Table 8: The Prompt for QA task evaluation inherited from Video-ChatGPT(Muhammad Maaz & Khan, 2023).

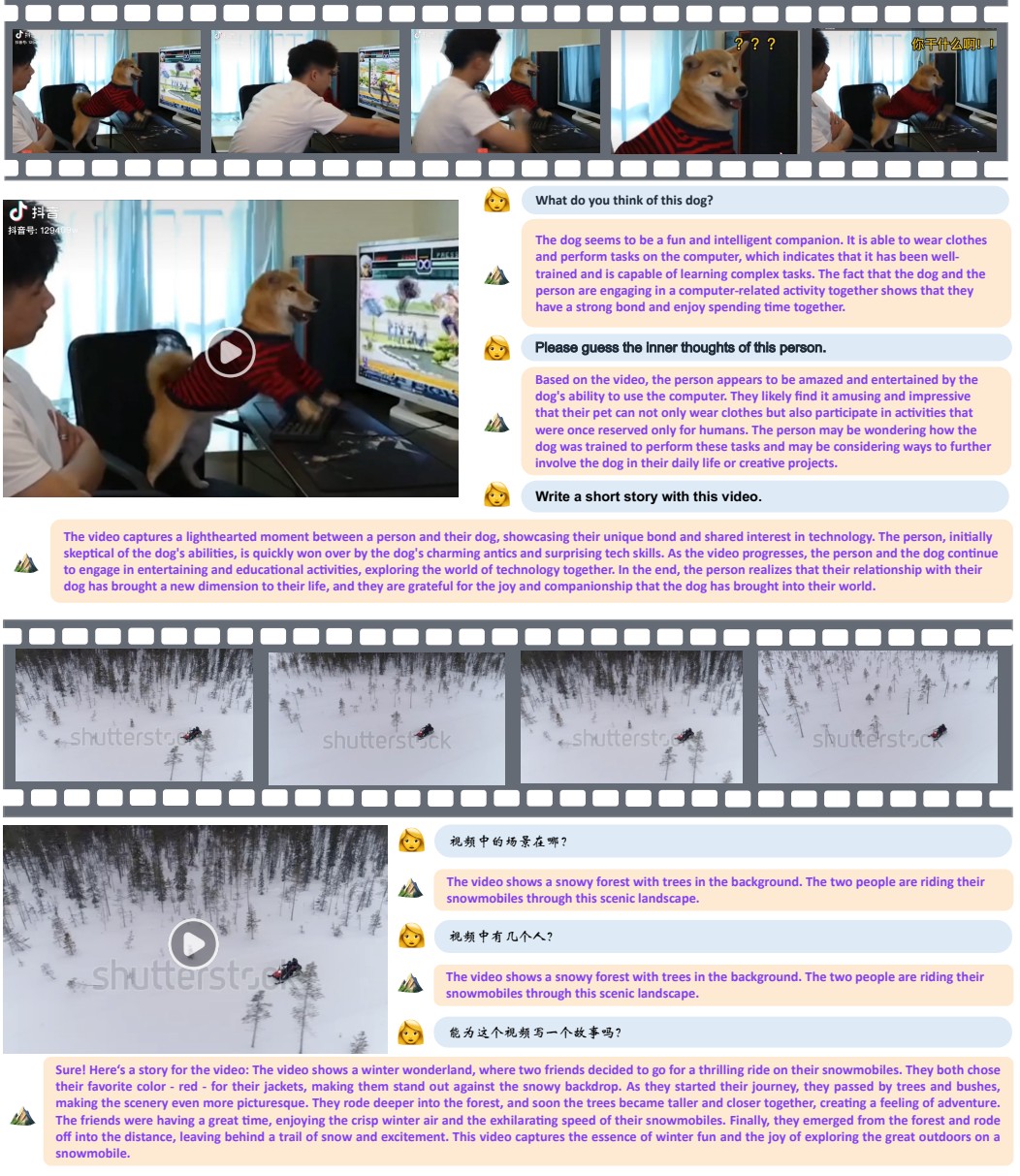

Figure 8: Capture the main content of the video, do the causal inference, and identify objects and relationships in the video. Chinese prompts are also well understood.

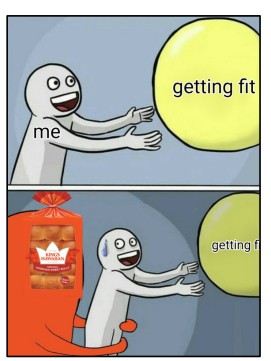

**Correct Answer:**
- Meme poster is trying to get healthy but feels temptation eating unhealthy foods.
- Meme poster wants to get healthy but keeps eating fattening King's Hawaiian rolls.
 - The meme poster isn't able get fit because of his food habits.
- Poster knows white bread will derail health goal.

*Valley*⛰: The meme poster is trying to convey a playful and humorous approach to the idea of getting into shape or working out. It features a cartoon character experiencing two contrasting situations: happily receiving a giant yellow ball (which likely symbolizes an enjoyable fitness activity) in the first frame, and then worryingly getting hugged by a giant bag of dinner rolls in the second frame. The two scenes suggest that the character is struggling with their fitness journey, and the humorous scenario is meant to resonate with the audience in a lighthearted manner.

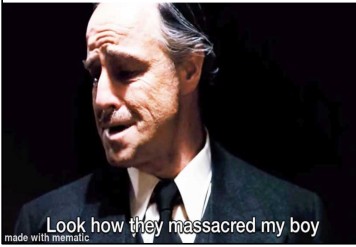

**Correct Answer:**
**-** The poster gets angry that another college student who shares things with him has damaged his non-stick pan.
- Meme poster is sad when someone uses their non-stick pan wrongly.
- The meme poster feels sad for the people who uses spatula on the non stick pan.

*Valley*⛰: The meme poster aims to convey a sense of anger, frustration or disgust by using a metal faucet as a visual metaphor for a situation that causes unhappiness or annoyance. The phrase "When someone uses a metal spatula on your non-stick pan" suggests that the problem may be persistent or difficult to resolve. The meme poster is likely expressing its dissatisfaction or disappointment with the situation, using the imagery of a spoiled pan to represent the concept.

Figure 9: Two valley examples in the Meme-cap benchmark. The blue box represents the human-annotated understanding of the image, and the green box represents valley's understanding of the image.

