# OpenReview forum: "Valley: Video Assistant with Large Language model Enhanced abilitY"
_ICLR.cc/2024/Conference — Submitted to ICLR 2024_

### Official Review · Reviewer_Vdhg · 2023-10-23

**Soundness:** 3 good
**Presentation:** 3 good
**Contribution:** 3 good
**Rating:** 5
**Confidence:** 4

**Summary:**

This paper introduces Valley for comprehending video, image, and language within a general framework. Valley is developed by combining a large language model, a temporal modeling module, a visual encoder, and a projection module, and is trained on a video instruction dataset using a two-stage tuning procedure. Experiments show that Valley has the potential to be a highly effective video assistant.

**Strengths:**

1. The paper is well-written and easy to follow.
2. The authors collect a video instruction-following dataset using ChatGPT with diverse types of tasks and categories, which can be helpful for instruction tuning VideoLLMs.
3. The propsed Valley is able to comprehend video, image, and language within a general framework.

**Weaknesses:**

1. The authors mention that they collect videos from Jukinmedia2 that provides videos with wide detailed descriptions, but do not clarify where the descriptions come from. They claim that the instrcution data constructed in this way will not bring about the illusion of object level. However, since the data is constructed by ChatGPT, which is prone to hallucination, this claim may not be entirely accurate.
2. The authors evaluate their model on general video question answering tasks, but on domain-specific image understanding benchmarks,  which seems illogical to me. As the authors claim that Vally can comprehend video, image, and language within a general framework, common image question answering datasets such as VQAv2 and GQA should be evaluated.

**Questions:**

Please see Weaknesses.

**Details Of Ethics Concerns:**

In my opinion, no ethics review are needed.

---

> ### Author Response · Authors · 2023-11-23
> **Rebuttal to Reviewer Vdhg**
>
> We are thankful for your meticulous review and valuable recommendations. It is heartening to see that you recognize the merits of this work. Our responses to your queries are provided below:
>
> ### Q1: About descriptions of collected video instruction dataset from Jukinmedia2 and less hallucination of this dataset.
>
> **A**: See details in global response
>
> ### Q2: Lack of domain-specific image understanding benchmarks
>
> **A**: We acknowledge that our focus is primarily on video understanding, which is why we have fewer experiments specifically related to images. However, we want to emphasize that Valley, in terms of its structure, is capable of comprehending both images and videos. The image case analyses presented in the paper demonstrate this capability. Also, we have evaluated our valley model on LLaVA benchmark，see details in global response. Results have demonstrated its effectiveness on image understanding.
>
> We appreciate your suggestion to include benchmarks such as VQAv2 and GQA in our future work.

---

### Official Review · Reviewer_1s3m · 2023-11-01

**Soundness:** 2 fair
**Presentation:** 2 fair
**Contribution:** 2 fair
**Rating:** 3
**Confidence:** 4

**Summary:**

This paper introduces Valley, a multimodal foundational model designed to comprehend video, image, and language within a unified framework. The authors have constructed a video instruction dataset using ChatGPT, which aids in the creation of task-oriented conversation data covering a wide range of tasks. A two-stage tuning procedure is adopted for model training. Both qualitative and quantitative experiments reveal that Valley holds promise as an effective video assistant, capable of simplifying complex video understanding scenarios.

**Strengths:**

- This paper gathers a 73k video-based instruction dataset with the help of ChatGPT. This is somewhat larger than the instruction datasets used in previous methods (e.g., VideoChat uses 11K video instruction data). Judging from the experimental results provided by the authors, the quality of this dataset appears to be quite good.
- Experiments show that Valley excels in visual question answering and captioning, demonstrating optimal performance, strong zero-shot capability. It also generates content with fewer hallucinations compared to similar models.

**Weaknesses:**

- In terms of instruction dataset construction, there seems to be a lack of innovation and comparative experiments. It appears that a higher-quality data source was simply used to collect data, and then common methods were employed to construct the instruction dataset. This was combined with instruction datasets from previous methods to obtain a larger instruction dataset. The decent performance achieved by this method in quantitative analysis may reflect the quality of the instruction dataset to some extent. However, there has been no comparison of the quality of the instruction dataset collected in this paper with that of previous datasets under fair conditions (such as equal data volume).
- In terms of the model methodology, there is a lack of innovation and comparative experiments. This paper attempts three temporal modeling strategies for video input, but these are very basic methods that have been widely used in tasks such as action recognition. There are no effective improvements aimed at enhancing conversational capabilities. Furthermore, the comparison with other methods like VideoChat is not detailed enough. More comparisons are needed (e.g. performance comparison under
the same dataset or same backbone) to demonstrate that the temporal modeling and other modules used in this paper have superior performance and efficiency.

**Questions:**

1. More fair experimental comparisons are needed to demonstrate that the instruction dataset collected in this paper is of higher quality than previous instruction datasets.
2. More fair experimental comparisons are needed to demonstrate that the methodology proposed in this paper performs better in terms of performance or efficiency compared to previous methods.
3. Are there any technical contributions that I have overlooked? In my view, the biggest difference between Valley and previous methods is the instruction dataset used.

---

> ### Author Response · Authors · 2023-11-23
> **Rebuttal to Reviewer 1s3m**
>
> We are appreciative of your comprehensive review and beneficial suggestions. It is encouraging to know that you acknowledge the strengths of this work. Our answers to your questions are as follows:
> ### Q1&Q2: Regarding the fairness of our proposed Valley and dataset
>
> **A**: We re-trained on the valley-v1 structure using only videochat data (videochat-instruct-11k), and the experimental result on three datasets (MSVD, MSRVTT, and ActivityNet) are detailed in global response. We can see that under the same amount of instruction data (11k), Valley performs better than the videochat model of the same size (13b). It shows that we use simple linear layers to align video and text modalities is effective. And we prove that splice VIT's "[CLS]" token for temporal representation performs well.
>
> And through experiments, we also prove the advantages and disadvantages of different timing modeling methods (valley-v1, valley-v2, valley-v3) under the same amount data, which has guiding significance for future research.
>
>
>
> ### Q3: Regarding the novelty of our proposed Valley
>
> **A**:
> 1. The prevalent works, such as VideoChat, Video LLaMA, adopt Q-former to bridge the textual and visual modality, while we turn to the simpler linear projection layer.
> 2. Similar concurrent works such as Video-ChatGPT also use the linear projection layer, but its use mean pooling on both spatial and temporal tokens for video modeling, which is sub-optimal in building video assistant. We design three temporal modeling methods, and experimentally verified the superiority of valley-v3 in temporal understanding compared to other baselines.
> 3. We collect and design a large video instruction-following dataset with the assistance of ChatGPT. Experiments demonstrate the quality of our proposed dataset. We have released this dataset 'Valley-702K', and there are already works beneficial from our proposed dataset such as *Video-LLaVA*[1]
>
> ### Reference
>
> [1] Lin B, Zhu B, Ye Y, et al. Video-LLaVA: Learning United Visual Representation by Alignment Before Projection[J]. arXiv preprint arXiv:2311.10122, 2023.

---

### Official Review · Reviewer_AvQ5 · 2023-11-01

**Soundness:** 3 good
**Presentation:** 3 good
**Contribution:** 3 good
**Rating:** 6
**Confidence:** 5

**Summary:**

This paper presents a new video-based model using instruction-tuning. The key is to collect a large number of videos with detailed captions. Given the collected video-instruct data as well as the publicly available image-instruct data, they conduct a two-stage training method: (1) train the projection layer and (2) train both the projection layer and LLM. Experimental results show promising improvements on multiple public benchmarks.

**Strengths:**

- This is a simple and effective method. The paper is well-written and easy to follow.
- In my humble opinion, this work could be one of the first to explore instruction tuning in the video domain.
- Strong results on multiple benchmarks.
- The constructed dataset should be a valuable resource to the community.

**Weaknesses:**

- While the data collection pipeline is well-formulated, this method requires very high-quality training data.  Gathering high-quality video instruction data remains very challenging when aiming for large-scale training. This prohibits very large-scale training to significantly boosting the model quality, especially when it comes to the video domain where the video data is often sparse and requires a very large number of training data.
-  The direct integration of vision transformers and LLMs may encounter obstacles, especially when dealing with lengthy videos. While the method was tested on some benchmarks, the videos tested in this paper are often very short (only a few seconds). There is still a long distance to real-world video applications.
- It is also difficult to ensure no hallucination in the instruction tuning data.

**Questions:**

see weakness

---

> ### Author Response · Authors · 2023-11-23
> **Rebuttal to Reviewer AvQ5**
>
> We are grateful for your thorough review and helpful recommendations. And we are encouraged that you appreciate the strengths of this work. Our responses to your questions are as follows:
>
> ### Q1：About high-quality video instruction data.
>
> **A**: We completely agree with your viewpoint and acknowledge the importance of high-quality video instruction data for our method. We are also well aware that video data in the domain is often sparse and requires a substantial amount of high-quality training data.  Therefore, in our current research, we have made significant efforts to gather the available high-quality data from open websites and design instruction and conversations to train the model. Based on our knowledge, it is of the largest scale in existing video instruction dataset. And the results prove its "high quality". We hope the data will be helpful to the community and call for researchers to seek additional resources and collaborations to obtain more high-quality video data from the real world.
>
> ### Q2:  Obstacles between vision transformers and LLMs and long distance to real-world video applications due to the length.
>
> **A**: Thank you for your insightful comments regarding the misalignment between the Vision Transformer (ViT) and the Language and Vision Model (LLM). To address this issue, ViT from CLIP which has undergone text-to-vision alignment training is selected and we have pretrained a projection layer with large amount of video-text pairs and image-text pairs to effectively align the representations. Regarding the evaluation datasets, we selected them based on wide usage in video understanding research. However, we acknowledge the importance of considering a diverse range of evaluation datasets, especially lengthy videos, and will take your suggestion into account for future research.
>
> ### Q3: Difficult to ensure no hallucination in the instruction tuning data.
>
> **A**: See details in Global response, other works have used our dataset, e.g. *VideoLLaVA*[1]
>
> ### Reference:
>
> [1] Lin B, Zhu B, Ye Y, et al. Video-LLaVA: Learning United Visual Representation by Alignment Before Projection[J]. arXiv preprint arXiv:2311.10122, 2023.

---

> > ### Comment · Reviewer_AvQ5 · 2023-11-23
> > **Response**
> >
> > Thank you for providing answers to my questions. Agree with most of the replies which are largely aligned with my initial review, and this is also my reason for initially voting for a score 6 (weak accept).
> >
> > Regarding Q2, my question is not about the alignment, but more about the model design to potentially process various video lengths (i.e., ranging from a few seconds to hours). I wonder if the authors have some possible remedies to address this challenge.  In fact, this question is relevant to the technical novelty concerns raised by fellow reviewers.
> >
> > On the other hand, I totally agree with using the widely used benchmarks. However, in the era of LLMs/LMMs, these models have demonstrated advanced capabilities that can directly address real-world problems. In my humble opinion, the current eval benchmark may not clearly reflect the full capability of the advanced LLM/LMM-based models. In other words, evaluating with the selected benchmark (e.g., with very short videos) may limit the scope of this work.

---

> > > ### Author Response · Authors · 2023-11-23
> > > **Replying to AvQ5**
> > >
> > > Thank you for your quick reply.
> > >
> > > For the challenge of video length variation, we designed valley-v3. The valley-v3 we designed supports video input of varying lengths (video sampling based on fps). In practical applications, for longer videos, the sampling fps value can be reduced.
> > >
> > > And for your second concern, your understanding of this issue is very accurate. Indeed, evaluating the capabilities of multimodal large models is a challenging problem. Existing benchmark tests cannot fully cover all scenarios and cannot exhaustively capture all aspects of perceptual analysis. Therefore, it is a relatively reasonable approach to manually collect video data from the real world and construct evaluation sets from multiple dimensions to assess the understanding and reasoning abilities of large models. However, this method may introduce subjectivity and may not be widely accepted by everyone, and it requires a significant amount of human resources. Currently, it is difficult for us to achieve this, but we can work together to continuously improve and refine evaluation methods.

---

### Official Review · Reviewer_rHRB · 2023-11-02

**Soundness:** 3 good
**Presentation:** 2 fair
**Contribution:** 2 fair
**Rating:** 5
**Confidence:** 3

**Summary:**

This work proposes large video-language model Valley which consists of an LLM, a temporal modeling module, a visual encoder, and a cross-modality projection module. This work also constructs a video instruction dataset using the ChatGPT to obtain conversational data for multi-shot captions, long video descriptions, action recognition, and causal relationship inference. Valley is evaluated on different benchmarks including MSVD-QA, MSRVTT-QA, Meme-Cap and Science-QA.

**Strengths:**

* Valley achieves the state-of-the-art performance of multiple video QA benchmarks MSVD-QA, MSRVTT-QA and ActivityNet-QA.
* Valley collects a dataset of 100k videos with detailed caption and plans to release the dataset which will benefit the research community.

**Weaknesses:**

* It is not clear what is the technical novelty of the proposed method Valley. Throughout the introduction, related works, and method sections there is not statement that explains the technical difference distinct from the existing video-language models.
* No ablation is provided other than the temporal modeling modules (v1, v2, v3), which also makes it difficult to judge what technical component mainly contributes to the performance.
* Among the temporal modeling modules, what is the unique advantage of the version-2 with linear layers over v1 and v3? There seems no benchmark where the v2 performs the best and so it is not clear why v2 is proposed as one of the main methods.
* There is no implementation details. It seems not easy to reproduce the result based on what is provided in the main manuscript.

**Questions:**

* While the dataset collection is one of the main contributions, there seems no license information about the used video data (https://www.jukinmedia.com/).
* What is the model size and runtime comparison with other state-of-the-art methods (VideoChat, Video LLaMA, Video-ChatGPT)?

**Details Of Ethics Concerns:**

This work collects and plans to release a dataset with 100K videos from https://www.jukinmedia.com/, and corresponding text annotations using the ChatGPT.

---

> ### Author Response · Authors · 2023-11-23
> **Rebuttal to Reviewer rHRB**
>
> Thank you for your valuable feedback. Below, we show you the point-to-point responses, which we hope can address your concerns.
>
> ### Q1: Regarding the novelty of our proposed Valley
>
> **A**: We propose employing a simple linear projection layer as a bridge between the video and text modalities, while similar works mostly adopt the complex Q-former structure. Besides, we specifically design three temporal modeling modules  for improving video understanding.
>
> ### Q2: Regarding the unique advantage of the version-2 with linear layers over v1 and v3
>
> **A**: For improving v1, v2 introduces  a learnable linear layer to learn the temporal importance score of a certain frame. In order to more fully model the temporal variation representation of these spatial tokens, we derived the v3 based on Transformer. In light of experiments, the performance of V2 is basically between V1 and V3, such as the MSRVTT-QA and the video-based text generation benchmark provided by Video-ChatGPT. Refer to Table 1 and Table 2.
>
> ### Q3: Regarding the implementation details.
>
> **A**: We will release all the source code to facilitate the reproduction of the results in the paper.
>
> ### Q4: About model size and inference time.
>
> **A**: Model size of VideoChatGPT and LLaMA-adaptor is 7B, and others are all 13B. We did not perform the inference of all these models and the evaluation metrics are claimed from the papers.
>
> **For more details of the novelty and ablations, please refer to our global response above.**

---

### Author Response · Authors · 2023-11-23
**Global Response**

Thank you very much for valuable opinions from all reviewers. We retrained and experimented on valley-v1 with the same amount of data as videochat. We provide the following unified responses to the questions that all reviewers are concerned about:

### Novelty

1. The prevalent works, such as VideoChat, Video LLaMA, adopt Q-former to bridge the textual and visual modality, while we turn to the simpler linear projection layer.

2. Similar concurrent works such as Video-ChatGPT also use the linear projection layer, but its use mean pooling on both spatial and temporal tokens for video modeling, which is sub-optimal in building video assistant. We design three temporal modeling methods, and experimentally verified the superiority of valley-v3 in temporal understanding compared to other baselines which has guiding significance for future research.

3. We collect and design a large video instruction-following dataset with the assistance of ChatGPT. Experiments demonstrate the quality of our proposed dataset. We have released this dataset 'Valley-702K' for modal alignment and 'Valley-instruct-73k' for instruction tuning, and there are already works beneficial from our proposed dataset such as *Video-LLaVA*.

### Video instruction data and hallucinations

|  	Model    |     Instruct Tuning Data | MSVD Acc | MSRVTT Acc | ActivityNet Acc  |
|:----------:|:------:|:---:|:------:|:-----:|
|  videochat |  videochat-instruct-11k | 56.3 | 45.0 | 26.5 |
|  Valley-v1  |  videochat-instruct-11k | 57.1 | 45.3 | 35.9 |
|  Valley-v1  |  videochat-instruct-11k + Valley-instruct-73k | 65.4 | 45.7  | 42.9 |
|  Valley-v2  |  videochat-instruct-11k + Valley-instruct-73k | 59.1 | 49.9  | 32.5 |
|  Valley-v3  |   videochat-instruct-11k + Valley-instruct-73k | 60.5 | 51.1 | 45.1 |

1. The second line of table above shows the experimental results of Valley-v1 trained only on the instruction dataset released by VideoChat. We can see that with the same amount of data, the performance of valley-v1 is significantly better than videochat using Q-former as the modal alignment module. This proves the effectiveness of the valley-v1 structure. By comparing our experimental results of lines 2 and 3, after adding the video instruction fine-tuning data we collected, the performance of Valley-v1 increased rapidly. This proves that dataset we collect is of higher quality.

2. To foster community development, we will release the collected datasets.

3. It is exciting to note that recent work, such as Video-LLaVA, has adopted our dataset and achieved good results.


### Additional experiments on image understanding benchmark

|  	Model    | Conv | Detail | Complex | All   |
|:----------:|:------:|:---:|:------:|:-----:|
|  LLaVA |  83.1 | 75.3 | 96.5  | 85.1 |
|  InstructBLIP|  81.9 | 68.0 | 91.2  | 80.5 |
|  MiniGPT4|  65.0 | 67.3 | 76.6 | 69.7 |
|  Valley-v1  |  84.7 | 70.5 | 90.0  | 81.7 |
|  Valley-v2 |  72.2 | 56.9 | 93.0  | 73.4 |
|  Valley-v3 |  72.0 | 59.6 | 88.3  | 74.0 |

The results show our valley model is also capable of comprehending images well. Our model is primarily on video understanding, which is why we have fewer experiments specifically related to images.

---

### Meta-Review · Area_Chair_y8XY · 2023-12-09

**Metareview:**

The paper received overall negative ratings (3, 5, 5, 6). The reviewers acknowledged the strong empirical performance and the new dataset as the strengths of the submission. The major concerns are about unjustified claims regarding hallucination issues and the inconclusive discussion on the scale vs. quality of the dataset with regards to the performance improvement. The authors provided a detailed rebuttal, addressing some of the concerns. However, the reviewers explicitly stated that their core concerns have not been adequately addressed, and as a result, one reviewer lowered their rating and others maintained their initial ratings. Overall, the paper has some merits, but the remaining concerns of the reviewers make it difficult to recommend acceptance.

**Justification For Why Not Higher Score:**

The major concerns still remain even after rebuttal.

**Justification For Why Not Lower Score:**

N/A

---

### Decision · Program_Chairs · 2024-01-16

Reject